# Effects of Irrigation and Nitrogen Application Rates on Protein and Amino Acid Content of Sunflower Seed Kernels

**Liang Feng [1], Weiping Li [1,\*], Qingxiao Shi [1], Sha Zhao [1], Yunfeng Hao [2], Hanjiang Liu [2] and Haibin Shi [1,\*]**

[1] Water Conservancy and Civil Engineering College, Inner Mongolia Agricultural University, Hohhot 010018, China; fengliang@emails.imau.edu.cn (L.F.); sqx_18654507577@163.com (Q.S.); zhaosha@emails.imau.edu.cn (S.Z.)

[2] City Agricultural and Animal Husbandry Research Institute of Bayannur, Bayannur 015000, China; hyf63310@163.com (Y.H.); liuhanjiang666@163.com (H.L.)

\* Correspondence: nmnd_liwp@163.com (W.L.); shb@imau.edu.cn (H.S.)

**Abstract:** Sunflower seeds are rich in oil and protein. In a two-year field experiment in Hetao district, Inner Mongolia, we evaluated the effects of irrigation and N fertilizer on protein and amino acid content of sunflower seeds (Kang Di T562 variety). Irrigation water and N fertilizer were applied at the bud to blossom stage and at three levels: water at 90, 45, and 0 mm; N fertilizer at 104, 52, and 0 kg hm$^{-2}$. There were nine treatments with three replicates for a total of 27 plots arranged randomly in blocks. In combination with environmental conditions, water, nitrogen, and the interaction between the two had significant effects on total protein, total amino acids, total ammonia, and key amino acids: glutamic acid, arginine, aspartic acid, glycine, and valine. Application of N fertilizer at the bud stage promoted protein synthesis in seed kernels at the ripening stage. We found a significant positive correlation between N fertilizer and the five main amino acids. With increasing irrigation, total amino acid content and the content of each of the key amino acids individually first increased but then decreased. Overall, a water deficit improved protein content in seed kernels.

**Keywords:** sunflower; quality; N fertilizer; water





## 1. Introduction

Sunflower is the third largest oil crop and the fourth largest vegetable oil crop in the world [1]. Mass production in a limited number of countries, with two-thirds of production concentrated in Europe, Russia, and Ukraine, provides more than three-quarters of the total trade in sunflower seed meal and sunflower seed oil [2]; other major producers are Argentina, China, the United States, etc. [1]. Sunflower seed kernels are rich in oil [3], protein [4], and amino acids [5,6]. Fortification/supplementation of food with plant proteins is important to ensure good nutrition in many regions; sunflowers represent an important and cheap source of plant protein for this application [7].

To produce crops with the maximum number of amino acids and high levels of protein, plants must receive nutrients, particularly nitrogen [8]. Crops cannot take up nitrogen directly from the air and must absorb it in different forms from the soil. The nitrogen use efficiency of plants depends on levels in the soil and on the rate of accumulation before flowering when it is redistributed to the fruit and seeds [9]. Water stress is the most important environmental variable affecting plant growth, development, and seed production. Effective irrigation and fertilization can double seed production [10]. The protein content and amino acid composition of sunflower seed kernels are not only affected by nitrogen fertilizer and irrigation, but also by genetic variation, soil fertility, sunshine, disease, and planting time [8].

Most previous studies on protein and amino acid levels in sunflower seed kernels considered single factors (e.g., irrigation or nitrogen) or single factors in combination with other agronomic measures. For example, Wu et al. only evaluated the effect of

fertilizer application, and showed that treatment with N and P together achieved the highest protein content per seed, whereas N, P, and K together achieved the highest total protein output (i.e., higher overall yield) [11]. Another study showed that nitrogen application could results in high yield and high protein content, but that the interaction between genotype and N application had no significant effect on protein content [12]. Another study showed that the protein content of sunflower seed kernels increased with increasing nitrogen application and decreased with increasing phosphorus application [13]. Gül et al. reported that the yield and quality of sunflower are affected by different nitrogen rates; the results showed that the highest protein content (27.2%) was observed with the 12 kg ha$^{-1}$ nitrogen treatment [14]. Munir et al. found that different application amounts of organic and inorganic fertilizers had significant effects on the protein content of sunflower seeds. The highest protein content was obtained under the treatment of 50-75-50 NPK kg ha$^{-1}$ with poultry fertilizer of 8 t ha$^{-1}$ [15]. Compared with organic fertilizers or inorganic fertilizers alone, combination fertilizers significantly increased the oil content and protein content of sunflower seeds; the protein content was highest under 50% farm manure [16]. Wang et al. studied the contents of proteins and amino acids in rape and sunflower pollen, and used the essential amino acid requirements published by the Food and Agriculture Organization (FAO) and the World Health Organization (WHO) in 1973 and the the National Research Council(NRC)in 1980 and their mutual ratios to evaluate the proteins in sunflower pollen. The results showed that the eight essential amino acids in sunflower pollen were balanced [17].

Zhang et al. studied the effects of altitude, latitude, and temperature on the protein and amino acid contents of sunflower seed kernel, and the results showed that the contents of protein and amino acid in sunflower seed kernel were higher at low altitude and the temperature difference was small [18].

Fernandez et al. analyzed the metabolomics characteristics of sunflower leaves, and using the minimum set of metabolic markers to distinguish genotypes or stress levels. The results showed that, based on statistical analysis, a limited number of markers are more accurately able to separate WW (well-watered) and DS (drought-stressed) samples than previously published physiological data [19]. Hocking studied the sulfur and nitrogen of sunflower seed yield and quality of two factors; the results showed that sulfur deficiency with unrestricted N application rate will affect the amino acid composition of seed, that with sufficient nitrogen, cysteine and methionine contents in plant seeds were reduced by 30%, and that the content of arginine of well-fed plants increased by 34% [20]. Ali et al. found that the increase in nitrogen level led to the stable growth of yield, protein content, and linoleic acid, whereas oleic acid content and oleic acid percentage showed a negative correlation over two years [21]. Steer et al. found that with the increase in nitrogen content in the dry weight of seeds, the concentration of amino acids also increased, and the increase ratio of arginine and glutamate was higher than that of the total increase ratio of amino acids [22]. Özer et al. examined the response of sunflower growth and yield composition to nitrogen fertilizer under irrigation [23]. Jami et al. evaluated the agronomic characteristics and seed quality of sunflower in response to different regimes of nitrogen, irrigation, and zeolite. The results showed that under no water stress, the application of zeolite could result in the highest oil and protein contents of grains the application of chemical fertilizer increased the protein content of grain, and organic fertilizer increased the oil content of grain [24]. Li et al. focused on the effects of N fertilizer on the quality of oil from sunflower seeds under the same irrigation regime. When a basic N fertilization application (68 kg ha$^{-1}$) was used, topdressing N applications during budding and flowering did not significantly increase crude fat content or linoleic acid content. However, topdressing N applications during budding and flowering promoted the synthesis of protein, palmitic acid, and oleic acid in the oil from sunflower seeds [25]. We previously studied the effects of different nitrogen application rates on the protein, crude fat, and fatty acid components of sunflower. The amino acid composition was not studied. On the basis of this research, further research was conducted; therefore, in this study of the canopy varieties, planting density, harvest

seed kernel parts, and under the premise of harvest, we explored the influence of water and nitrogen on the sunflower seed kernel protein and amino acids, revealing the response law of sunflower seed kernel protein content and amino acid composition to water and nitrogen, providing technical support for the quality development of the sunflower industry.

## 2. Material and Methods

### 2.1. Description of the Field Site

The field trial was conducted at Shahaoqu Experimental Station (40°55′37.43″ N, 107°09′55.88″ E, 1056 m above sea level), which is in the Jiefangzha irrigation area of Hetao Irrigation District, Inner Mongolia, from May to September 2016 and May to September 2017. This area has an arid to semi-arid continental climate. Meteorological data (max. and min. temperature, relative humidity, dew point temperature, wind speed, atmospheric pressure, solar radiation, and rainfall) were obtained from automatic weather stations (HOBO) located in the test areas during the trial (Table 1).

**Table 1.** Monthly meteorological data during the crop growth period.

| Year | Month | Maximum Temp (°C) | Minimum Temp (°C) | Relative Humidity (%) | Dew Point (°C) | Wind Speed (m/s) | Air Pressure (kPa) | Solar Radiation (W/m$^2$) | Rain (mm) |
|---|---|---|---|---|---|---|---|---|---|
| | May | 33.40 | 1.90 | 47.71 | 4.75 | 0.83 | 89.29 | 247.06 | 10.00 |
| | June | 35.70 | 10.70 | 58.05 | 12.52 | 0.61 | 89.00 | 272.31 | 27.80 |
| 2016 | July | 38.00 | 14.50 | 68.45 | 17.80 | 0.39 | 88.85 | 252.09 | 18.40 |
| | Aug. | 36.60 | 9.30 | 69.77 | 17.57 | 0.32 | 89.26 | 208.12 | 40.00 |
| | Sept. | 31.70 | 4.20 | 62.48 | 10.56 | 0.32 | 119.48 | 197.69 | 4.80 |
| | May | 35.10 | 0.30 | 32.71 | 0.81 | 0.96 | 89.65 | 265.49 | 2.40 |
| | June | 36.60 | 9.80 | 51.43 | 10.98 | 0.44 | 93.68 | 284.11 | 29.32 |
| 2017 | July | 35.60 | 13.10 | 61.76 | 16.69 | 0.27 | 89.14 | 251.65 | 10.40 |
| | Aug. | 33.80 | 8.60 | 56.79 | 12.61 | 0.20 | 89.65 | 246.28 | 5.40 |
| | Sept. | 32.90 | -0.60 | 48.39 | 6.85 | 0.25 | 96.50 | 208.57 | 3.00 |

Soil texture was determined using a laser particle size analyzer, and allocated to type according to the soil texture descriptions of the United States Department of Agriculture [26]. Soils at a depth of 0–20 and 40–60 cm were powdery loams; at a depth of 20–40 cm, it was powder. The available nitrogen, phosphorus, potassium, and organic matter content in soil at sowing were described as follows: at sowing, the available nitrogen content was 24.46, 20.38, and 16.85 mg·kg in the 0–20, 20–40, and 40–60 cm soil layers, respectively. At sowing, the available phosphorus content was 11.32, 6.05, and 5.94 mg·kg$^{-1}$ in the 0–20, 20–40, and 40–60 cm soil layers, respectively. At sowing, the available potassium content was 152.68, 131.45, and 119.52 mg·kg$^{-1}$ in the 0–20, 20–40, and 40–60 cm soil layers, respectively. At sowing, the organic matter content was 17.12, 14.59, and 7.03 g·kg$^{-1}$ in 0–20, 20–40, and 40–60 cm soil layers, respectively.

### 2.2. Experimental Design of Field Trial

The high-quality hybrid sunflower, KangDi T562, was selected. We adopted a two-factor randomized block design, and the study factors were: the amount of irrigation (three levels), and the amount of nitrogen (as urea) applied as a top dressing (three levels) at the bud stage. A base fertilizer was applied to all plots (diammonium phosphate; 18% N, equivalent to 68 kg hm$^{-2}$) before the plants were transplanted into the plots. The rates at which the treatment topdressing of urea (containing 46.4% N) were 104 kg hm$^{-2}$ (denoted as $N_{104}$), 52 kg·hm$^{-2}$ (denoted as $N_{52}$), and a zero application (denoted as $N_0$). Irrigation was applied by surface spraying using border irrigation methods, measured accurately using a water meter in the squaring stage. The three irrigation levels were 90 mm for high water level ($W_{90}$), 50% of the high-water level (45 mm; $W_{45}$), and no irrigation (0 mm, $W_0$). Fertilizer and water were applied together in different combination to create a total of nine

treatments (Table 2) and there were three replicate plots of each treatment. Thus, a total of 27 plots were arranged in three blocks (one replicate per treatment in each block; replicates arranged randomly).

**Table 2.** Experimental design.

| Treatment | Irrigation (mm) | Fertilizer (kg/hm) |
|---|---|---|
| $W_0N_0$ | 0 | 0 |
| $W_0N_{52}$ | 0 | 52 |
| $W_0N_{104}$ | 0 | 104 |
| $W_{45}N_0$ | 45 | 0 |
| $W_{45}N_{52}$ | 45 | 52 |
| $W_{45}N_{104}$ | 45 | 104 |
| $W_{90}N_0$ | 90 | 0 |
| $W_{90}N_{52}$ | 90 | 52 |
| $W_{90}N_{104}$ | 90 | 104 |

Each replicate experimental plot covered an area of $4 \times 8$ m ($32$ m$^2$) and was surrounded by impermeable plastic film to a depth of 1 m to prevent soil moisture and nutrients seeping between replicates of different treatments. Sunflower seeds were planted in nursery beds on 30 May 2016 and 1 June 2017. Once the plants achieved the 2-leaf stage, they were transplanted into the plots; three plants were placed into each of 26 planting holes per plot to achieve 1 plant per hole after thinning. There were eight rows of plants per plot; plants were 30–35 cm apart from each other, and the inter-row spacing was 60–65 cm to give 26 plants per plot. The planting density of each plot was consistent. Field management measures, such as weeding and pest control, were consistent with local farming practices.

*2.3. Collecting Plant Samples*

Five representative plants with the same flowering time were selected from 26 sunflower plants in each plot. After ripening, 50 seeds of each plant were extracted from the outermost three or four circles (the position was relatively large and full) to avoid the influence of different positions of the flower disc on protein content and amino acid composition of sunflower kernel oil. After mixing the number of seeds picked in each plot, three samples of seeds were taken for testing the kernel quality.

First, sunflower seeds were crushed for testing by a grinding machine. Protein content was determined using a kjeltec$^{TM}$ 8100 (FOSS, Beijing, China), in accordance with national standard GB 5009.5-2016. We placed a 1.5 g sample into the digestive tube, and added medicine (0.4 g copper sulfate, 6 g potassium sulfate, 20 mL sulfuric acid) for digestion. When the liquid in the digestive tube was green and transparent, we removed and cooled it, adding 50 mL of water, and automatically adding liquid. Distillation and titration were performed, and we record titration data on the kjeltec$^{TM}$ 8100 analyzer, which was the protein content. Amino acid composition was determined by a L-8900 amino acid analyzer (Hitachi High Technologies America, Inc., Dallas, TX, USA) in accordance with national standard GB 5009.124-2016. The sunflower seeds were crushed by a grinding machine and hydrolyzed into free amino acids by hydrochloric acid. After separation by an ion exchange column, the color reaction was produced with ninhydrin solution, and the content of amino acids was determined by the amino acid analyzer at 21 °C and a relative humidity of 33%.

*2.4. Statistical Analysis*

Experimental data are expressed as mean ± SD (standard deviation). The significance of differences in protein content and amino acid components in sunflower seed kernels produced under different fertilizer and irrigation treatments were determined by analysis of variance (two-way ANOVA) in SPSS22.0 (IBM, Armonk, NY, USA). When a parameter for any of the experimental groups was significant, Duncan's multiple range test was used. Differences were considered to be significant at $p < 0.05$ and highly significant at

$p < 0.01$. We also performed correlation analysis for the amino acid components and used Euclidian-based system clustering analysis for classification.

## 3. Results and Discussion

### 3.1. Effect of Irrigation and Nitrogen and the Interaction between the Tw, on Protein Content of Sunflower Seed Kernels

We found a significant effect of irrigation ($p < 0.05$) and N fertilizer use ($p < 0.05$) on the protein content of sunflower seed kernels and a significant interaction ($p < 0.05$) (Table 3).

**Table 3.** Results of ANOVA on the effect of irrigation and nitrogen application on protein content of sunflower seed kernels.

|  | Treatment | SS | df | MS | F | *p*-Value |
|---|---|---|---|---|---|---|
|  | Irrigation | 89.830 | 2 | 44.915 | 135.502 | 0.000 ** |
|  | Nitrogen | 291.101 | 2 | 145.550 | 439.104 | 0.000 ** |
| Protein | i × n | 26.222 | 4 | 6.555 | 19.777 | 0.000 ** |
|  | Error | 5.966 | 18 | 0.331 |  |  |
|  | Total | 413.119 | 26 |  |  |  |

** Significant at the 0.01 level; SS, standard deviation square; MS, mean square; i, irrigation; n, nitrogen.df, degree of freedom.

In the no irrigation treatments ($W_0$), we observed a significant difference in protein content between $N_0$, $N_{52}$, and $N_{104}$ nitrogen application levels in both years ($p < 0.05$). This indicated that protein content of seed kernels increased with increasing nitrogen in the absence of irrigation. At the $W_{45}$ level of irrigation, we also found a significant difference in protein content amongst all three nitrogen application levels in 2016 ($p < 0.05$), whereas in 2017, there were significant differences between the $N_0$ and $N_{104}$ treatments ($p < 0.05$) and between the $N_{52}$ and $N_{104}$ treatments ($p < 0.05$). In general, this showed that the protein content of seed kernels increased with increasing nitrogen application, At the $W_{90}$ level of irrigation, significant differences in protein content were identified between $N_0$ and $N_{52}$ treatments and between $N_0$ and $N_{104}$ treatments in both years. Again, protein content increased with increasing nitrogen application. Overall, considering mean values, protein content showed an increasing trend with increasing nitrogen application at all irrigation levels; in 2016 and 2017, protein content was highest in $W_0N_{104}$, at 25.3% and 24.3%, respectively. The reason for this is that nitrogen is the main component of protein and nucleic acids; therefore, nitrogen fertilizer can promote the synthesis of amino acids in seed kernel proteins [27]. However, increasing levels of irrigation had a negative effect on seed kernel protein synthesis: as the level of irrigation increased, seed kernel protein content showed a downward trend. In both years, we found a significant interaction between the effect of irrigation and nitrogen application on seed kernel protein content ($p < 0.05$; Figure 1).

The results showed that the protein content of sunflower seed kernels increased with increasing nitrogen application, which is consistent with the findings of previous studies [28–30]. Nitrogen is usually a limiting factor for sunflower growth [9,13,23,31–33], and different nitrogen fertilizers are required at different stages of reproduction [34]. Blamey et al. reported that the protein content of sunflower seeds in each season was significantly increased by applying nitrogen fertilizer [13]. Shoghi-Kalkhoran et al. found that when applying farmyard manure, chemical fertilizer, or a 50:50 combination of the two that the combination had the highest protein content [16]. These earlier studies focused on different levels and combinations of fertilizers on protein content, whereas we considered the effects of different fertilizer levels in combination with different levels of irrigation on sunflower seed protein content. Under no irrigation treatments, seed kernel protein content increased with increasing nitrogen application. However, water stress is considered to be a key factor influencing the growth, development, and spatial distribution of crops [7]. The level of irrigation can significantly impact crop growth and quality. Oraki et al. showed

that protein content increased with decreasing levels of irrigation [35]. Jalilian et al. found that a water deficit increased the protein content of sunflower seeds [36]. Our results are consistent with these previous studies, and showed that different combinations of water and nitrogen have significant effects on the protein content of sunflower seeds.

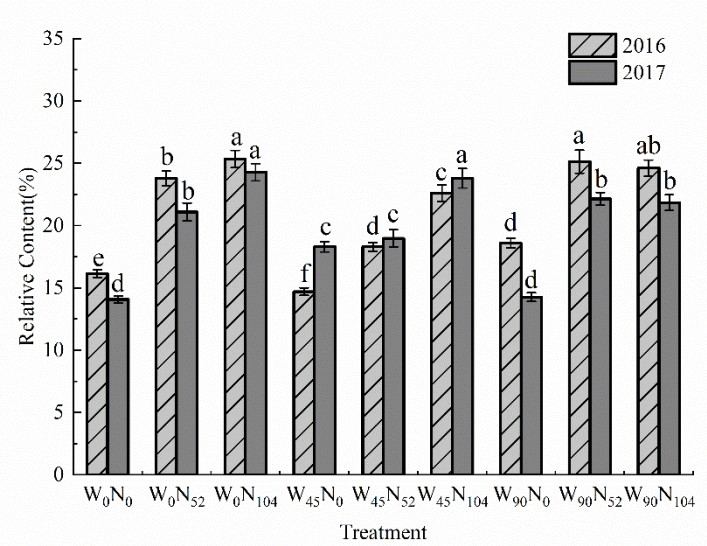

**Figure 1.** Influence of irrigation, nitrogen, and the interaction between the two on protein content of sunflower seed kernels (means ± SD; different lowercase letters indicate that there are significant differences in nine treatments at the 5% level in 2016 and 2017, respectively).

### 3.2. Effects of Irrigation and Nitrogen and the Interaction between the Two on Total Amino Acids, and Key Amino Acids Specifically in Sunflower Seed Kernels

In this study, a total of 17 major amino acid components in the kernels of oil sunflower seeds were tested for and all were present. To clarify the spatial correlation amongst the 17 amino acids, we used a systematic method based on Euclidean distance using cluster analysis on the data from samples, and a systematic clustering tree was obtained (Figure 2).

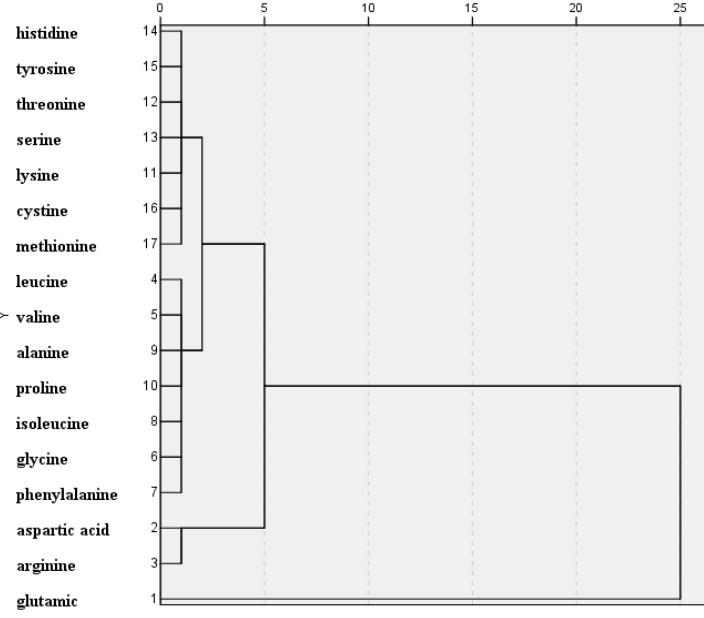

**Figure 2.** Clustering of amino acids identified in sunflower seed kernels.

When the threshold value was 1, the 17 amino acids clustered into four categories: histidine, lysine, tyrosine, threonine, serine, cystine, and methionine; leucine, valine, alanine, proline, isoleucine, glycine, and phenylalanine; aspartic acid and arginine grouped; and glutamic acid. When the threshold value was >2 and <5, the 17 amino acids clustered into three categories: glutamic acid; aspartic acid and arginine; and the remaining 14 in the third category. When the threshold value was >5, the 17 amino acids cluster into two categories: glutamic acid in one category, and the remaining 16 indicators in the other.

According to the test results, glutamate, aspartic acid, arginine, glycine, and valine were relatively high amongst the 17 amino acids from sunflower seed kernels. The correlation analysis of these five main amino acid components is shown in Table 4.

**Table 4.** Correlation coefficient of main amino acid components in sunflower seed kernels.

|  | Glutamic Acid | Aspartic Acid | Arginine | Valine | Glycine |
|---|---|---|---|---|---|
| Glutamic acid | 1 |  |  |  |  |
| Aspartic acid | 0.990 ** | 1 |  |  |  |
| Arginine | 0.990 ** | 0.990 ** | 1 |  |  |
| Valine | 0.990 ** | 0.990 ** | 0.980 ** | 1 |  |
| Glycine | 0.970 ** | 0.980 ** | 0.970 ** | 0.990 ** | 1 |

**: Correlation is significant at the 0.01 levels (2-tailed), respectively.

Differences in the total quantity of amino acids (excluding ammonia) and the quantity of each of the five main amino acids (glutamic acid, aspartic acid, arginine, glycine, and valine) in the sunflower seed kernels sampled from different treatments in 2016 to 2017 were analyzed by ANOVA. The two main factors were amount of irrigation and quantity of nitrogen fertilizer. We also considered the interaction between irrigation and nitrogen (Table 5). The level of irrigation had a greater effect than nitrogen level on the total amino acid content and on each of the five main amino acids individually ($p < 0.05$). From Duncan's multiple comparisons of means, we found no significant difference between those marked with the same letters; lower case letters represent $p < 0.05$ in Figures 3–9.

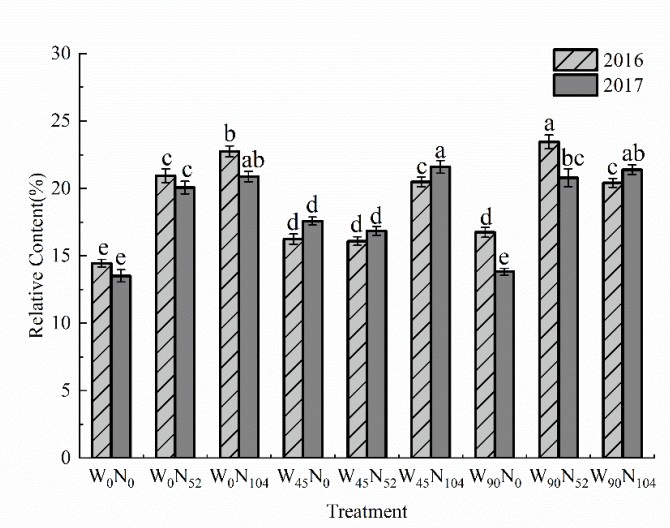

**Figure 3.** Influence of irrigation, nitrogen, and the interaction between the two on the total amino acid content of sunflower seed kernels (mean ± SD; different lowercase letters indicate that there are significant differences in nine treatments at the 5% level in 2016 and 2017, respectively.).

**Table 5.** Results of ANOVA on the effect of irrigation and nitrogen application on amino acid composition.

| Amino acid | Treatment | SS | df | MS | F | *p*-Value |
|---|---|---|---|---|---|---|
| Total amino acid | Irrigation | 1.630 | 2 | 0.820 | 4.450 | 0.027 * |
| | Nitrogen | 187.330 | 2 | 93.670 | 511.940 | 0.000 ** |
| | i × n | 56.078 | 4 | 14.020 | 76.630 | 0.000 ** |
| | Error | 3.290 | 18 | 0.180 | | |
| | Total | 248.330 | 26 | | | |
| Glutamic acid | Irrigation | 2.596 | 2 | 1.298 | 82.288 | 0.000 ** |
| | Nitrogen | 12.480 | 2 | 6.240 | 395.596 | 0.000 ** |
| | i × n | 5.451 | 4 | 1.363 | 86.389 | 0.000 ** |
| | Error | 0.284 | 18 | 0.016 | | |
| | Total | 20.811 | 26 | | | |
| Aspartic acid | Irrigation | 0.245 | 2 | 0.122 | 61.562 | 0.000 ** |
| | Nitrogen | 1.341 | 2 | 0.671 | 337.368 | 0.000 ** |
| | i × n | 0.683 | 4 | 0.171 | 85.904 | 0.000 ** |
| | Error | 0.036 | 18 | 0.002 | | |
| | Total | 2.305 | 26 | | | |
| Arginine | Irrigation | 0.501 | 2 | 0.250 | 98.151 | 0.000 ** |
| | Nitrogen | 2.041 | 2 | 1.021 | 400.004 | 0.000 ** |
| | i × n | 0.818 | 4 | 0.205 | 80.150 | 0.000 ** |
| | Error | 0.046 | 18 | 0.003 | | |
| | Total | 3.406 | 26 | | | |
| Valine | Irrigation | 0.083 | 2 | 0.041 | 52.749 | 0.000 ** |
| | Nitrogen | 0.498 | 2 | 0.249 | 317.099 | 0.000 ** |
| | i ×n | 0.278 | 4 | 0.070 | 88.498 | 0.000 ** |
| | Error | 0.014 | 18 | 0.001 | | |
| | Total | 0.874 | 26 | | | |
| Glycine | Irrigation | 0.040 | 2 | 0.020 | 33.874 | 0.000 ** |
| | Nitrogen | 0.326 | 2 | 0.163 | 274.887 | 0.000 ** |
| | i × n | 0.225 | 4 | 0.056 | 95.055 | 0.000 ** |
| | Error | 0.011 | 18 | 0.001 | | |
| | Total | 0.602 | 26 | | | |

** Significant at the 0.01 level; * significant at the 0.05 level.

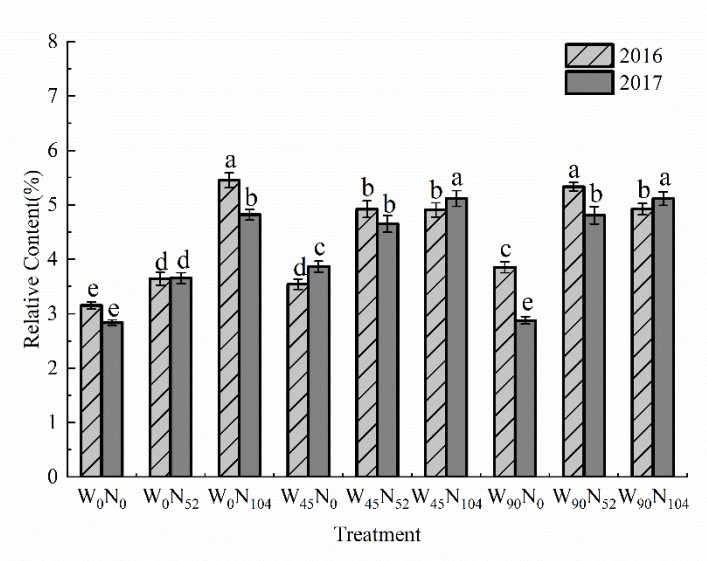

**Figure 4.** Effect of irrigation, nitrogen, and the interaction between the two on glutamic acid content of sunflower seed kernels (mean ± SD; different lowercase letters indicate that there are significant differences in nine treatments at the 5% level in 2016 and 2017, respectively).

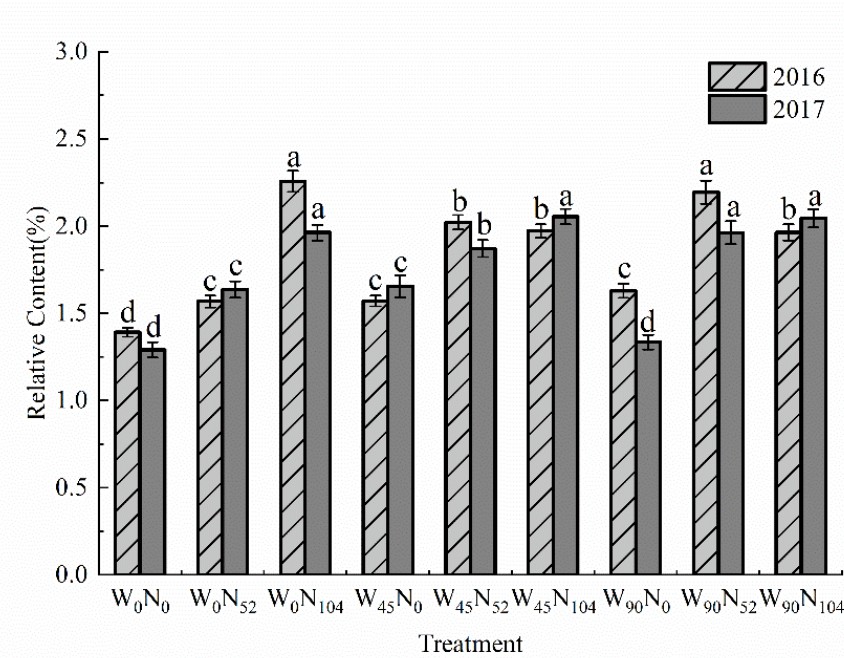

**Figure 5.** Effects of irrigation, nitrogen, and the interaction between the two on aspartic acid content of sunflower seed kernels (mean ± SD; different lowercase letters indicate that there are significant differences in nine treatments at the 5% level in 2016 and 2017, respectively).

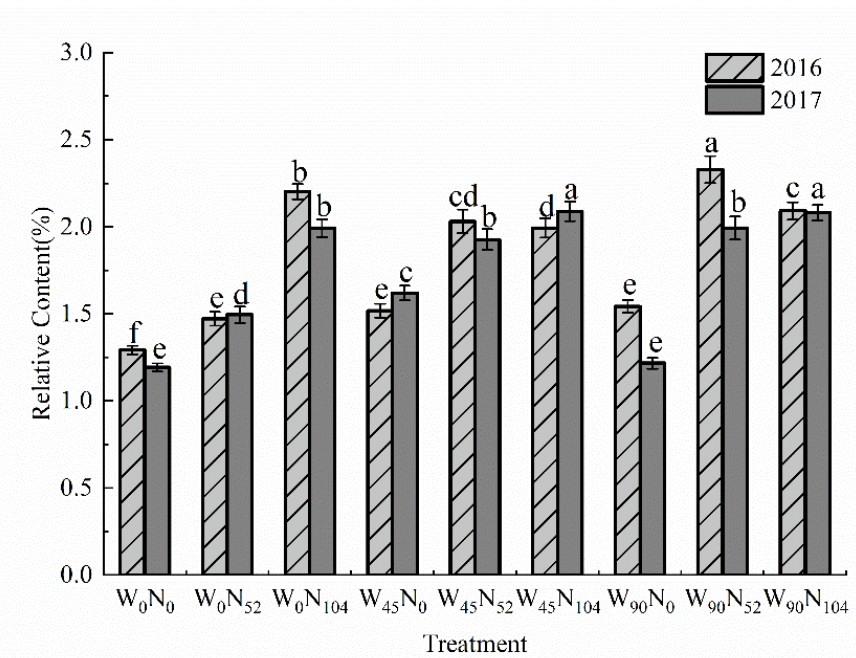

**Figure 6.** Effects of irrigation, nitrogen application, and the interaction between them on the arginine content of sunflower seed kernels (mean ± SD; different lowercase letters indicate that there are significant differences in nine treatments at the 5% level in 2016 and 2017, respectively).

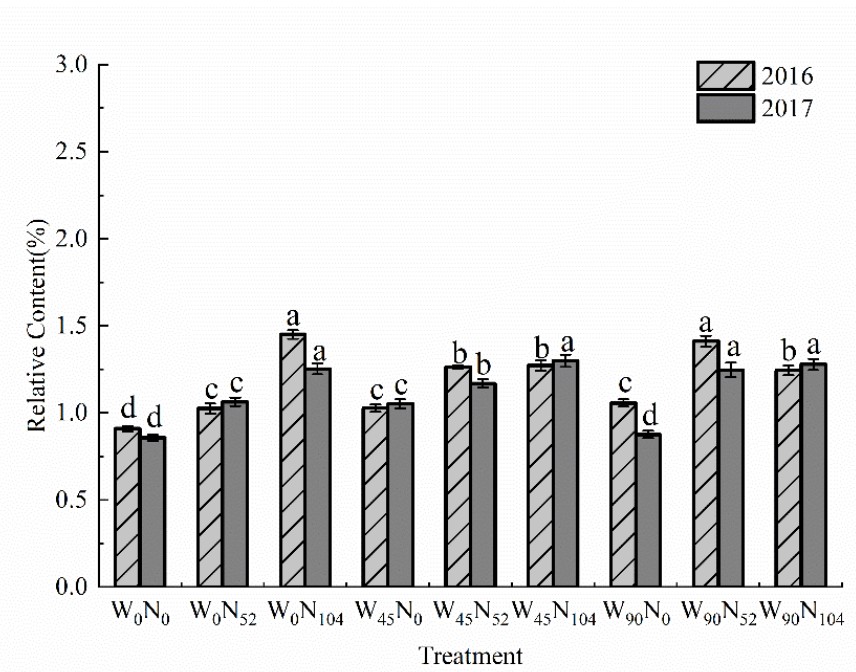

**Figure 7.** Effects of irrigation, nitrogen, and the interaction between the two on valine content of sunflower seed kernels (mean ± SD; different lowercase letters indicate that there are significant differences in nine treatments at the 5% level in 2016 and 2017, respectively).

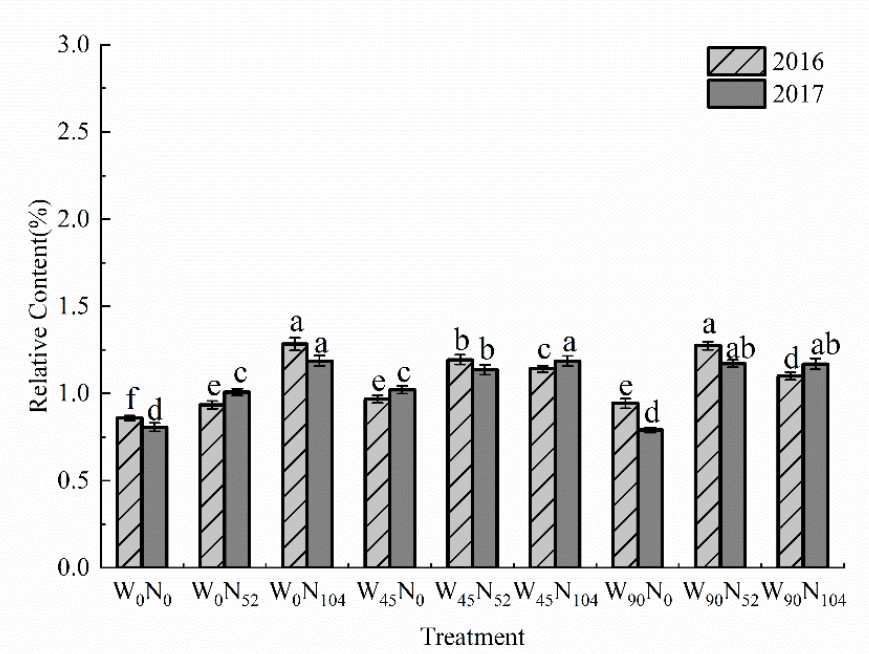

**Figure 8.** Effects of irrigation, nitrogen, and the interaction between the two on glycine content in sunflower seed kernels (mean ± SD; different lowercase letters indicate that there are significant differences in nine treatments at the 5% level in 2016 and 2017, respectively).

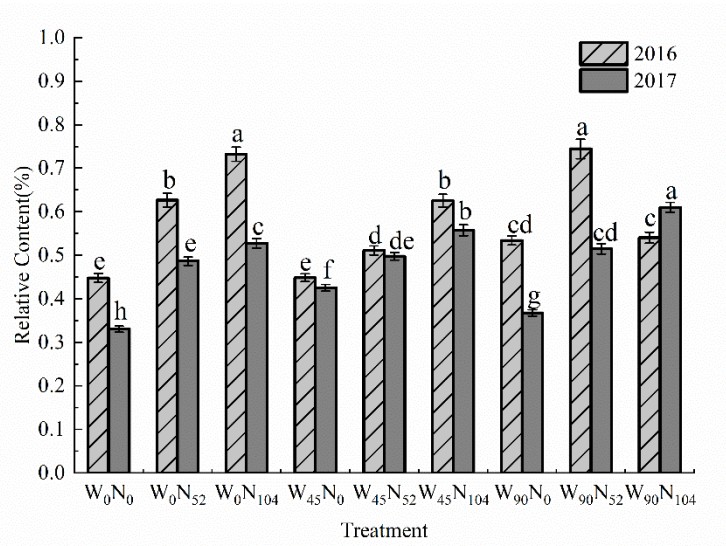

**Figure 9.** Influence of irrigation and nitrogen and the interaction between the two on ammonia content of sunflower seed kernels (mean ± SD; different lowercase letters indicate that there are significant differences in nine treatments at the 5% level in 2016 and 2017, respectively).

In the no irrigation treatments ($W_0$), we found significant differences in the total amino acid content of kernels amongst all three nitrogen treatments in 2016 ($p < 0.05$); in 2017, there were significant differences between the $N_0$ and $N_{52}$ treatments and between the $N_0$ and $N_{104}$ treatments ($p < 0.05$), but no significant difference between the $N_{52}$ and $N_{104}$ treatments ($p > 0.05$). In general, the total amino acid content increased with increasing nitrogen application. At the $W_{45}$ level of irrigation, results were consistent in both years: there was no significant difference in total amino acid content between the $N_0$ and $N_{52}$ treatments ($p > 0.05$), but there were significant differences in the total amino acid content between $N_0$ and $N_{104}$ treatments and between $N_{52}$ and $N_{104}$ treatments ($p < 0.05$). Overall, we found a trend of increasing total amino acid content with increasing nitrogen application. At the $W_{90}$ level of irrigation, there was a significant difference in the total content of amino acids amongst all three nitrogen application levels in 2016 ($p < 0.05$); total amino acid content increased first and then decreased. In 2017, significant differences were identified in the total amino acid content between $N_0$ and $N_{52}$ and between $N_0$ and $N_{104}$ nitrogen treatments ($p < 0.05$), but no significant difference between $N_{52}$ and $N_{104}$ treatments ($p > 0.05$).

Overall, for both years, total amino acid content decreased as the irrigation amount increased from 45 to 90 mm at the $N_{104}$ nitrogen application level. Therefore, at the $N_{104}$ level of nitrogen, 45 mm irrigation benefits seed kernel amino acid synthesis, but when irrigation increased to 90 mm, the excess water was detrimental to amino acid synthesis. Natural rainfall during the experimental period was significantly higher in 2016 (101 mm) than in 2017 (50.52 mm). In particular, rainfall from the budding period to the flowering period was 58.40 mm in 2016, and only 15.80 mm in 2017. These rainfall levels were sufficient to meet the normal growth and development needs of the crop in 2016, meaning that the maximum total amino acid content (22.74%) could be achieved in the absence of irrigation at the highest nitrogen application rate ($W_0N_{104}$). However, in 2017, 45 mm irrigation was required at the same nitrogen application rate ($W_{45}N_{104}$) to achieve a similar total amino acid content (21.59%).

In the no irrigation treatments ($W_0$), results for both years showed that there was a significant difference in glutamic acid content amongst all nitrogen treatments ($p < 0.05$); glutamic acid content increased with increasing nitrogen application. At the $W_{45}$ level of irrigation, there was no significant difference in glutamic acid content between $N_{52}$ and $N_{104}$ treatments in 2016 ($p > 0.05$), but there were significant differences in glutamic content between the $N_{104}$ treatment and the other two treatments ($N_0$, $N_{52}$) ($p < 0.05$). In 2017, there

were significant differences in glutamic acid content amongst all three nitrogen application levels ($p < 0.05$). At the $W_{90}$ irrigation level, we observed significant differences in glutamic acid content amongst all nitrogen treatments in both years ($p < 0.05$). In 2016, glutamic acid content increased first and then decreased with increasing nitrogen application. In 2017, glutamic content increased with increasing nitrogen application but, compared with the $W_{45}$ level of irrigation, glutamic acid content decreased by 0.01%. Overall, at all irrigation levels, increasing the amount of nitrogen applied was conducive to the synthesis of glutamic acid. As water hydrolyzes nitrogen in the soil and makes it available for plants, appropriate irrigation is conducive to the synthesis of glutamic acid. However, with higher irrigation (90 mm), glutamic acid content declined because the dissolved is leached from the soil by the excess water, limiting the synthesis of glutamate. For the reasons described previously, differences in natural rainfall in the two years meant that glutamic acid content was the highest in the $W_0N_{104}$ treatment (5.45%) in 2016 and in the $W_{45}N_{104}$ treatment (5.12%) in 2017.

In the no irrigation treatment ($W_0$), there were significant differences in aspartic acid content amongst all three nitrogen treatments in both years ($p < 0.05$). Overall, aspartic acid content increased with increasing nitrogen application. At the $W_{45}$ irrigation level, there were significant differences in aspartic acid content amongst all three nitrogen treatments in 2016 ($p < 0.05$), with the highest value for the year in the $W_0N_{104}$ treatment (2.26%), but there was no significant difference in aspartic acid content between the $N_{52}$ and the $N_{104}$ treatments ($p > 0.05$). In 2017, we found significant differences in aspartic acid content amongst all three nitrogen treatments, with the highest value for the year in the $W_0N_{104}$ treatment (2.06%). At the $W_{90}$ level, there were significant differences in aspartic acid content amongst all the nitrogen treatments in 2016; aspartic acid content first increased and then decreased. In 2017, significant differences were observed in aspartic acid content between $N_0$ and $N_{52}$ and between $N_0$ and $N_{104}$ treatments, but no significant difference between the $N_{52}$ and $N_{104}$ treatments; the trend was observed for increasing aspartic acid content with increasing nitrogen application. At all irrigation levels, the general trend was that increasing nitrogen application was beneficial for the synthesis of aspartic acid. At all nitrogen levels, 45 mm irrigation creased arginine content and was beneficial for the synthesis of aspartic acid. However, when irrigation increased to 90 mm, there was a downward trend in arginine content, likely because nitrogen dissolved in the water and was leached from the soil.

In the no irrigation treatments ($W_0$), we observed a significant difference in arginine content amongst all three nitrogen treatments in both years ($p < 0.05$). Overall, arginine content increased with increased nitrogen application, indicating increased nitrogen application was beneficial to the synthesis of arginine. At the $W_{45}$ irrigation level, we found no significant difference in arginine content between the $N_{52}$ and $N_{104}$ treatments in both years ($p > 0.05$) but there were significant differences in arginine content between the $N_0$ treatment and the other two nitrogen treatments ($p < 0.05$). Arginine content increased with increasing nitrogen in both years and was highest in the $W_{45}N_{104}$ treatment (2.09%) in 2017. At the $W_{90}$ irrigation level, there were significant differences in arginine content amongst the three nitrogen application levels in 2016 ($p < 0.05$). Arginine content first increased and then decreased. In 2017, there were significant differences in arginine content between the $N_0$ and $N_{52}$ treatments and between the $N_0$ and $N_{104}$ treatments ($p < 0.05$), but no significant difference between $N_{52}$ and $N_{104}$ treatments ($p > 0.05$). Overall, the trend was for arginine content to increase with increasing nitrogen application level. At all irrigation levels, the general trend was for increasing nitrogen application to benefit the synthesis of arginine. At all nitrogen levels, irrigation at levels of 45 mm increased arginine content and benefitted the synthesis of arginine. However, when irrigation increased to 90 mm, we observed a downward trend in arginine content, likely because nitrogen dissolved in the water and was leached from the soil.

In the no irrigation treatments ($W_0$), we found significant differences in the valine content amongst all three nitrogen treatments in both years ($p < 0.05$); valine content

increased with increasing nitrogen application in the absence of irrigation. At the $W_{45}$ irrigation level, there were significant differences in the valine content between the $N_0$ and $N_{52}$ treatment and between the $N_0$ and $N_{104}$ treatment in 2016 ($p < 0.05$), but none in the valine content between the $N_{52}$ and $N_{104}$ treatments ($p > 0.05$). In 2017, we noted a significant difference in valine content amongst all three nitrogen treatments ($p < 0.05$); overall, valine content increased with increasing nitrogen levels. At the $W_{90}$ irrigation level, valine content was significantly different amongst all three nitrogen treatments in 2016 ($p < 0.05$). In 2017, there were significant differences in the valine contents between the $N_0$ and $N_{52}$ treatment and between the $N_0$ and $N_{104}$ treatment ($p < 0.05$), but no significant difference in valine content between the $N_{52}$ and $N_{104}$ treatments ($p > 0.05$). Overall, at the $W_{90}$ irrigation level, valine content showed a trend of first increasing with nitrogen application and then decreasing in 2016; in 2017, it continuously increased with N application rate. From the perspective of mean valine values, valine content increased with increasing nitrogen application as increasing nitrogen application benefitted the synthesis of valine. However, when irrigation levels increased to 90 mm, glycine content declined at the highest nitrogen application rate because the water leached the nitrogen from the soil. For the reasons described previously, differences in natural rainfall in the two years meant that valine content was the highest in the $W_0N_{104}$ treatment (1.45%) in 2016 and in the $W_{45}N_{104}$ treatment (1.30%) in 2017.

In the no irrigation treatment ($W_0$), there were significant differences in the glycine contents amongst all three nitrogen treatments in both years ($p < 0.05$); glycine content increased with increasing nitrogen application. At the $W_{45}$ irrigation level, there were significant differences in the glycine contents between all three nitrogen treatments in 2016 ($p < 0.05$); in 2017, significant differences were observed in glycine content between $N_0$ and $N_{52}$ and between $N_0$ and $N_{104}$ ($p < 0.05$), but no significant difference between $N_{52}$ and $N_{104}$ treatments ($p > 0.05$). Overall, we found a trend for glycine content to increase with increasing nitrogen application rate. At the $W_{90}$ irrigation level, we identified significant differences in glycine content amongst all three nitrogen levels ($p < 0.05$); in 2017, it was similar, except there was no significant difference between $N_{52}$ and $N_{104}$ treatments ($p > 0.05$). From the perspective of mean glycine values, glycine content increased with increasing nitrogen application. However, when irrigation levels increased to 90 mm, glycine content declined at the highest nitrogen application rate because the water leached the nitrogen from the soil. For the reasons described previously, differences in the natural rainfall in the two years meant that glycine content was the highest in the $W_0N_{104}$ treatment (1.28%) in 2016, and in the $W_{45}N_{104}$ treatment (1.19%) in 2017.

The results were similar for arginine and aspartic acid; both had the highest values in the $W_0N_{104}$ treatment in 2016 and the $W_{45}N_{104}$ treatment in 2017 due to a positive correlation between arginine and aspartic acid (Table 4). In the cluster diagram, these two amino acids were significantly grouped together. We found significant differences in valine and glycine contents in relation to their interactions with irrigation and nitrogen application. The effect of nitrogen application on valine content was significantly stronger than the effect of irrigation level. Both valine and glycine had their highest levels in the $W_0N_{104}$ treatment in 2016 and the $W_{45}N_{104}$ treatment in 2017 due to a positive correlation between them (Table 4). In the cluster diagram, valine and glycine were classified into one category. Our findings showed that under different levels of irrigation, an increase in topdressings of nitrogen was associated with an increase in total amino acid content, and the contents of glutamic acid and arginine, which is consistent with the study of Steer et al. [22].

In total, 17 amino acids contributed to the total amino acid content; this did not include ammonia, which was present but analyzed separately. ANOVA showed that irrigation and nitrogen had a significant effect on the ammonia content of sunflower seed kernels, as did the interaction between them (Table 6). We also performed a multiple comparisons of means (Figure 9).

**Table 6.** ANOVA to determine the effect of irrigation and nitrogen application on ammonia in sunflower seed kernels.

| Amino Acid | Treatment | SS | df | MS | F | *p*-Value |
|---|---|---|---|---|---|---|
| | Irrigation | 0.040 | 2 | 0.017 | 89.005 | 0.000 ** |
| | Nitrogen | 0.141 | 2 | 0.070 | 362.454 | 0.000 ** |
| Ammonia | I × n | 0.117 | 4 | 0.029 | 150.255 | 0.000 ** |
| | Error | 0.004 | 18 | 0.000 | | |
| | Total | 0.296 | 26 | | | |

** Significant at the 0.01 level;

In the no irrigation treatments ($W_0$), we found significant differences in ammonia content between the $N_0$ and $N_{52}$ treatments and between the $N_0$ and $N_{104}$ treatments ($p < 0.05$), but not between the $N_{52}$ and $N_{104}$ treatments in both years ($p > 0.05$). Ammonia content increased with increasing nitrogen application. At the $W_{45}$ irrigation level, there was a significant difference in ammonia content amongst all three nitrogen treatments in both years ($p < 0.05$); ammonia content increased with increasing nitrogen application. At the $W_{90}$ irrigation level, ammonia content showed a trend of increasing first and then decreasing in 2016, achieving its highest level in the $W_{90}N_{45}$ treatment (0.74%). In 2017, significant differences were recorded in ammonia content amongst all three nitrogen treatment ($p < 0.05$); ammonia content increased with increasing nitrogen application, achieving its highest level in the $W_{90}N_{104}$ treatment (0.61%). From the perspective of mean values, ammonia content increased with increasing nitrogen application; therefore, nitrogen application favored ammonia synthesis in sunflower seed kernels.

To more clearly identify the impact of the different treatments of irrigation and fertilizer on the quality of sunflower seeds, principal component analysis was performed on all experimental data from 2016 to 2017 (Figure 10).

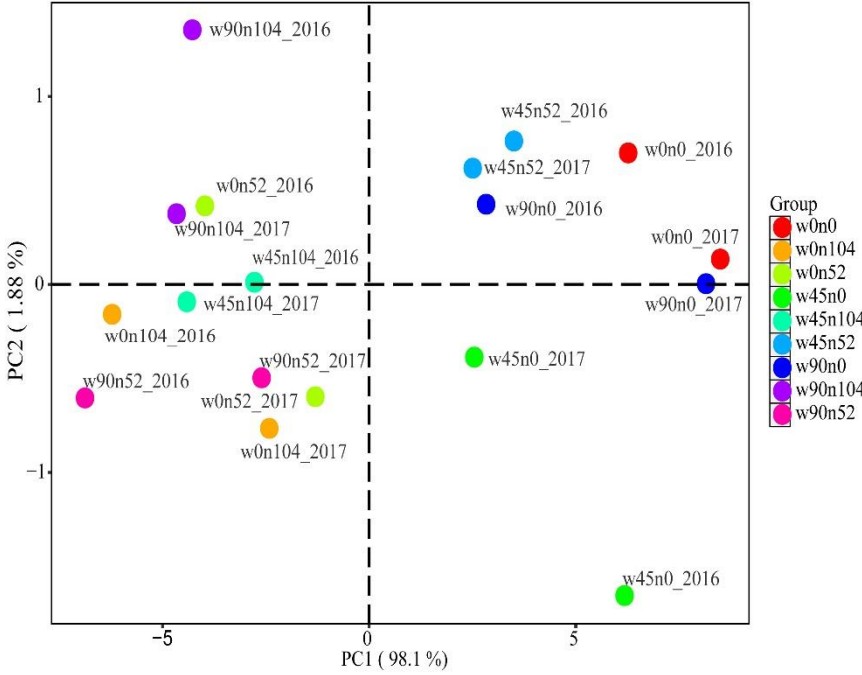

**Figure 10.** Principal component analysis of eight indicators on irrigation and fertilization amount in 2016 and 2017.

The interaction responses of the contents of protein, total amino acids, five main amino acids, and ammonia in sunflower seed kernel to water and nitrogen were the same in different years. For the above eight quality indicators, the difference was significant for different water and nitrogen treatments. The protein content of sunflower seed kernels

increased with increasing nitrogen application. Increasing nitrogen application was directly related to increased total amino acid content and the synthesis of the five key amino acids in sunflower seeds.

## 4. Conclusions

Different levels of irrigation and nitrogen application and their interactions had significant effects ($p < 0.05$) on protein content, amino acid components (glutamic, aspartic acid, arginine, glycine, and valine), and ammonia at budding stage of sunflower. Topdressing with nitrogen fertilizer during budding can promote protein synthesis in the seed kernel. In the absence of irrigation, protein content increased with increasing nitrogen in both years, achieving the highest levels in the $W_0N_{104}$ treatment at 25.34%, and 24.28% in 2016 and 2017, respectively. Increasing nitrogen helped improve seed kernel protein content. We observed a significant positive correlation between the five main amino acids in sunflower seed kernels and nitrogen application at the budding stage. Increasing nitrogen was directly related to increased total amino acid content and the synthesis of the five key amino acids in sunflower seed kernels during maturation.

**Author Contributions:** Conceptualization, L.F., W.L., H.S., Q.S., S.Z., H.L., and Y.H.; methodology, L.F.; software, L.F.; validation, L.F., W.L, H.S., Q.S., S.Z., H.L., and Y.H.; Data curation, L.F.; Writing—original draft preparation, L.F.; Writing—review and editing, L.F., W.L., and H.S. All authors have read and agreed to the published version of the manuscript.

**Funding:** This research was funded by the key project of the Natural Science Foundation of China, grant number 51539005; the National Nature Science Foundation of China, grant number 52069022; and the major project of Inner Mongolia Science and technology, grant number zdzx2018059.

**Institutional Review Board Statement:** Not applicable.

**Informed Consent Statement:** Not applicable.

**Data Availability Statement:** Not applicable.

**Conflicts of Interest:** The authors declare no conflict of interest.

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
