# Peer review of "Effects of Irrigation and Nitrogen Application Rates on Protein and Amino Acid Content of Sunflower Seed Kernels"

_water, doi:10.3390/w13010078_

Round 1

Reviewer 1 Report

In the manuscript entitled “Effects of Irrigation and Nitrogen Application Rates on Protein and Amino Acid Content of Sunflower Seed Kernels” the authors analyse the protein and amino acid content of Sunflower under combination with environmental conditions (water, nitrogen and the interaction WxI). They analysed global protein content and 17 major amino acid and showed several variations of content under the nine treatments.

I appreciate the experimental design of field trial and the methodology that seem to be very useful and powerful for this type of study (blocks arranged randomly, number of rep, number of seeds sampled / flowerhead, … ). But, I have many concerns that deserve to be considered by the authors in order to gain further clarity and impact.

  1. The introduction is not entirely clear to me, and I am confused by the choice of references.

- The first sentence L29 for example, defined the crop as important in subtropical and temperate regions but cite study done in Pakistan in 2012 [1] and not a review in the subject like: Pilorgé, E. (2020). Sunflower in the global vegetable oil system: situation, specificities and perspectives. OCL, 27, 34. DOI: 10.1051/ocl/2020028

- The second sentence L30 introduce the sunflower world production but cite a study to response to potassium [2] and not the USDA report for example (https://www.fas.usda.gov/data/oilseeds-world-markets-and-trade). Because this second sentence is false as mentioned in Pilorgé 2020 and USDA : Sunflower is produced at large scale in a limited number of countries, and two thirds of the production are concentrated in Europe, including Ukraine and Russia and the Trakya region of Turkey.

- L47 : You mentioned previous studies with different consideration of irrigation and nitrogen in single factor but cite only study on the effect of fertilizer. You could read and cited some recent articles that study drought stress like: Fernandez, O., Urrutia, M., Berton, T., Bernillon, S., Deborde, C., Jacob, D., ... & Langlade, N. B. (2019). Metabolomic characterization of sunflower leaf allows discriminating genotype groups or stress levels with a minimal set of metabolic markers. Metabolomics, 15(4), 56.

- Many recent references about your question are missing, need to be ad or verified, like :

  • Ali, A., & Ullah, S. (2012). Effect of nitrogen on achene protein, oil, fatty acid profile, and yield of sunflower hybrids. Chilean Journal of Agricultural Research, 72(4), 564.
  • Ozer, H., Polat, T., & Ozturk, E. (2004 AND NOT 2006). Response of irrigated sunflower (Helianthus annuus L.) hybrids to nitrogen fertilization: growth, yield and yield components. Plant Soil and Environment, 50(5), 205-211.
  • GÜL, V., & KARA, K. (2015). Effects of Different Nitrogen Doses on Yield and Quality Traits of Common Sunflower (Helianthus annuus L.). Turkish Journal of Field Crops, 20(2), 159-165.
  • Jami, M. G., Ghalavand, A., Modarres-Sanavy, S. A. M., & Mokhtassi-Bidgoli, A. (2018). Evaluation of agronomic characteristics and seed quality of sunflower in response to different regimes of nitrogen, irrigation and zeolite. Journal of Crops Improvement, 19(4).

- I wonder why the authors do not cite their own and recent research done in the same field :

  • - Li, W. P., Shi, H. B., Zhu, K., Zheng, Q., & Xu, Z. (2017). The quality of sunflower seed oil changes in response to nitrogen fertilizer. Agronomy Journal, 109(6), 2499-2507.
  1. Material and Methods section:

- L131: the Collecting Plant Samples part is correctly detailed but, the protein and amino acid extraction, analyzed… should be much more detailed and placed in a separate section.

- L140: The statistical analysis part is poorly described. We need to know if you used the 26 plants per plot or if they are on average and you test only the three replicates per treatment. I have the same concern about the 50 seeds sampled. Have you pooled them together or analyzed each one independently?

Your figures mention some letters, I think it is a Duncan's Multiple Range or a Tukey post hoc test but need to explain and describe the method and above all, the significance of the test used (0.01, 0.05, 0.1?). In all your figures, there are many overlapping SD bars, between years and treatments and in these cases they are not significantly different and the letters must be shared. In fact, this gap prevents me from thinking and trusting the presented and detailed results.

  1. Results:

There is no big difference in paternity between the 2 years of study which is very comforting. So I think a choice of the year can be made by the authors to clarify the article's message. The authors would only show the results of a single year and give the results of the second one in supplemental figures.

Also to help reading, the authors may be performed a PCA analysis with all the data to show more clearly the impact of the different treatments. Authors can also graphically display a correlation matrix of their data to aid in data interpretation. This type of analyses must be done to gain clarity and impact.

The y-axis for figure 5, 6, 7 and 8 must be adapted to the data, a max of 2 or 3 % may be applied for example.

Major comments:

- Generally, the writing of this manuscript should be improved.

- All the data must be available (all the protein content and amino acid composition) in additional file (Raw data) or in a public depository.

- I do not understand why the discussion part is so short. As mentioned in the instructions for authors of the Water journal, “This section may be combined with Results.” and may be a good way for the authors.

I am aware of the important analysis, process and rewriting work requested in this review but I give an absolutely positive assessment to the data produced and I think that the new version of this article will have more impact and visibility.

Reviewer 2 Report

The paper presents results of two years of experimental studies focused on influence of irrigation and N fertilization on protein and amino acids content in seeds of one of the most important economic plant – sunflower.

The results are presented clearly showing that increased N fertilization with no irrigation (W0N104) can highly enlarge protein content in the seeds of helianthus which is even higher then observed for well irrigated and well fertilized fields. In my opinion the observation is very important from economic and environmental point of view because water resources are becoming more limited in large areas of the globe. Helianthus is commonly known as a source of oil. The study of proteins and selected amino acids content in helianthus seeds is very important taking into account inevitable gradual replacement of animal proteins by plant proteins in human diet due to increasing human population.

Line 57 – ‘in organic’ - the translation is confusing

Lines 66-69 – especially the fragment ‘that the content of protein and amino acid in sunflower seed kernel was high with low altitude and small temperature difference’ is not clear.

Lines 78-84 – the whole, long sentence is very unclear

Line 451 – ‘fatty proteins’ - is it correct?

Round 2

Reviewer 1 Report

Thank you for carefully studying my comments, I appreciate the correction made. But I am still concerned about some comments I have already written.

- one of my major former comments:  All the data must be available (all the protein content and amino acid composition) in additional file (Raw data) or in a public depository.

- same question as my previous review: your figures mention letters from Duncan's Multiple Range test but we don’t know what you test by your two-way ANOVA. It is by years? By treatment? By treatment x years? In all your figures, there are many overlapping SD bars, between years and treatments and in these cases, they are not significantly different and the letters must be shared (and that is not the case). For example, figure 1, W0N52 / 2016 = 25 like W0N104 / 2017 and W45N104 / 2016 and 2017 but the first one has "b" letters and the other "a". Figure 2, W0N0 in 2016 and 2017 have the same % of relative content (=15) but with different letter of statistical test (respectively c and d). We see the same things for many treatments, figures … and this must be verified. This key question in statistics prevents me from trusting the presented and detailed results. This may be due to the amount of data you used to calculate each mean / SD that may be different from the data used for the statistical test.

- The legend of your PCA figure indicates eight indicators used but I count nine colors in your graph. Your PCA show 99.98 (98.1 + 1.88) of variance explained with only the two first components, that is huge ! The only way to interpret this is that your data is very strongly correlated. Once again, I believe that the dissemination of the data will also allow the reader to better understand these results and to trust them.

Round 3

Reviewer 1 Report

Thank you for carefully studying my last comments, I very much appreciate the correction made.